# Psychometric Validation of the Brazilian Portuguese Version of the Derriford Appearance Scale-24 (DAS-24) for People Living with HIV/AIDS

**DOI:** 10.3390/healthcare8040569

**Published:** 2020-12-17

**Authors:** Marcos Alberto Martins, Angela Nogueira Neves, Tim Moss, Walter Henrique Martins, Gerson Vilhena Pereira, Karina Viviani de Oliveira Pessôa, Mariliza Henrique da Silva, Luiz Carlos de Abreu

**Affiliations:** 1Laboratório de Delineamento de Estudos e Escrita Científica, Centro Univeristário Saúde ABC, Santo André, São Paulo 09060-870, Brazil; whenriquemartins@uol.com.br (W.H.M.); gerson_vilhena@uol.com.br (G.V.P.); luizcarlos@usp.br (L.C.d.A.); 2Hospital Samaritano de São Paulo—Cirurgia Plástica, São Paulo 01232-010, Brazil; 3Divisão de Pesquisa, Escola de Educação Física do Exército, Rio de Janeiro 22291-090, Brazil; angela.esefex@yahoo.com.br; 4Postgraduate Research, Faculty of Health and Applied Sciences, University of the West of England, Bristol BS16 1QY, UK; tim.moss@uwe.ac.uk; 5Programa Municipal de IST/AIDS e Hepatites Virais de São Bernardo do Campo, São Bernardo do Campo, São Paulo 04121-000, Brazil; karinaviviani@terra.com.br; 6Programa Estadual de IST/AIDS do Estado de São Paulo, São Paulo 04121-000, Brazil; mariliza.henrique@gmail.com; 7School of Medicine, University of Limerick, V94 T9PX Limerick, Ireland

**Keywords:** distress, questionnaire, HIV/AIDS, Brazil, body image

## Abstract

The changes in appearance of people living with HIV/AIDS (PLHA) interferes with how people around them react to their body, how social interactions take place, and how each person perceives and accepts their body. The definition of itself can be severely challenged when the body changes as a result of illness and the person does not look healthier anymore. People living with HIV/AIDS (PLHA) are an especially vulnerable group when it comes to “distress” and the psychosocial impact of appearance, yet the assessment of body image changes in these people was subjective in Brazil. The aim of this paper was to assess the psychometric properties of the Brazilian version of Derriford Appearance Scale 24 (DAS-24) for a sample of Brazilians living with HIV/AIDS. A sample of 400 patients were recruited from an HIV/AIDS ambulatory, aged between 18 and 78 years, of both sexes. The psychometric properties of DAS-24 were investigated while using confirmatory factor analysis (CFA), with unweighted least square estimation and listwise deletion for missing data. The adjustment of three structural models previously established for DAS-24 (single-factor, two-factor, and three-factor) was investigated. Evidences of construct validity—convergent and discriminant—and internal consistency—Cronbach’s alpha and construct reliability—were also generated for the measure model. The results showed that the one-factor model had the best adjustment, after eliminating items 8, 17, and 20, and accepting the covariance of errors between items 4 and 10; 9 and 23; 11 and 14; and, 14 and 22. Additionally, validity and reliability evidence were satisfactory for the model. The Brazilian Portuguese version of DAS-24 seems to be a psychometrically sound scale for measuring body image distress for people living with HIV/AIDS (PLHA).

## 1. Introduction

Body image is the mental representation of the body, and it is linked to an integrated brain organization [1]. It is singular, dynamic, with conscious and unconscious components, being influenced by sensory, developmental, and psychodynamic factors [2]. The body’s appearance and function interfere with the way that people around us react to our bodies, how social interactions take place, and how we perceive and accept our own bodies. The body image of people with changes in appearance and/or function emerges from this context [3,4].

Nowadays, HIV positive serology is no longer a death sentence. Improvements in antiretroviral therapy (ART) have resulted in reduced mortality rates, increased quality of life, and a reduced likelihood of transmission to others [5]. After the diagnosis, there are two important moments, in which substantial changes in body image are experienced: the moment of diagnosis *per se* and the moment when the symptoms of the disease become visible [6,7].

Despite the knowledge of HIV non-transmissibility in the presence of viral suppression with appropriate treatment [8], for some people, the diagnosis of HIV causes a change in the perception of the body barrier, which precisely determines the limits of the body and its identity. At this moment, the phenomenological state of a “guaranteed” body is broken [9]. Common representations of this moment include metaphors as “in danger”, “at risk”, “punishment”, and “contaminated” [10]. The body becomes something to be monitored and, on it, the virus is chronically marked, which is reflected in social interactions changes and the concept of itself [6].

In continuity, when the symptoms and visible signs of HIV chronicity begin to become evident, changes in social contact, beliefs and self-concept, and affectivity can be observed [6,11]. Regarding the physical aspects, lipodystrophy is the condition that registers the presence of HIV in the body. Lipodystrophies in people living with HIV/Aids (PLHA) are a heterogeneous group of adipose tissue disorders, which are characterized by bodily changes caused by the redistribution of fat, which is, the loss of fat in the extremities (face, arms, legs, and buttocks) and accumulation of fat in the central portion (abdomen, back-cervical spine, female breasts, and gynecomastia in men). This occurs due to several factors, such as: HIV infection, genetic factors, use of antiretrovirals, among others [12] HIV infection and HAART have been associated with the development of cardiovascular and metabolic complications, including lipodystrophy syndrome. In Brazil, the prevalence of lipodystrophy manifestation has ranged from 32.4 to 64%, in line with the average found in the literature, of 42% (SESSP, 2017), with an overall prevalence of at least one physical abnormality that is related to the poor distribution of body fat, estimated at about 50% after one year of TARV use [13] with an impact on financial life, social relationships, and love relationships [7].

The body image changes that accompany lipodystrophy lie largely in the impossibility of socially denying or hiding the infection, exposing the patient to someone else’s judgment [14]. When evidencing HIV infection, the patient begins to experience the stigmatization of the disease. PLHA report decreased social contacts, being touched, and being able to touch each other [15], including at the service health (Stringer et al., 2016). They also report feeling “abnormal”, less attractive, and sexually desirable [16]. They experience prejudice, discrimination, job loss, violence, and threats to their personal well-being, due to their serological status [17,18]. It is important to highlight that the changes experienced in body image, especially body dissatisfaction, are associated with low adherence to treatment and general psychological distress [19,20]. The internalization of the stigma of being seroreagent is associated with guilt and shame, states of depression, anxiety, helplessness, stress, and social isolation, in addition to affecting physical health and well-being [21,22,23]. Persons with lipodystrophy syndrome report that dealing with lipodystrophy may be more difficult than HIV itself, because a subjective experience with appearance may be more challenging than objective reality itself [24]. 

However, the diagnosis of HIV and subsequent lifestyle changes during treatment can lead to a positive resignification of life [25]. “Posttraumatic growth” refers to positive changes—i.e.: appreciation of life, spirituality, meaningful relationships, and self-confidence that can occur after experiencing a negative event. Because this reframing is an observed and possible fact for PLHA [26], maintaining a systematic assessment of interventions that are made in the treatment system is important for assessing their effectiveness in the patient’s overall health.

### The Present Study

Among the instruments that evaluate “distress” and body discomfort, the Derriford Appearance Scale (DAS) stands out [27]. DAS was originally designed by researchers in the United Kingdom to be an objective measure of the spectrum of psychological stress and body image dysfunction in the characteristic aspects of disfigurement, deformities, and aesthetic problems [28]. Its original formulation consists of 59 items (DAS-59), organized into five factors—awareness of general appearance; awareness of social appearance; sexual and bodily awareness of appearance; negative self-concept; and, awareness of easy appearance. DAS-59 has an introductory section that allows the subject to identify and describe which aspect of his appearance he is most sensitive or inhibited and this is referred to as his “reference” to the responses of the scale items. Fifty-seven items of the scale evaluate psychological distress and psychological dysfunctions, and two items evaluate physical distress and physical dysfunctions [28]. 

Because it was thought to be used in research and clinical practice, its psychometric studies extended to subjects of both sexes, of all age groups from 16 years, healthy or not. Attention should be paid to the specific audience to whom it is addressed: to people who have had their body parts disfigured and wait for reconstruction surgery or who are already under treatment [29].

Five years after the creation of the DAS-59, a short version with 24 items was proposed DAS-24 [27]. This short version was elaborated for clinical use and routine follow-up of people with these disorders [27]. The DAS-24 items were removed from the DAS-59, and their choice was given by both what they could observe—concerns and behaviors regarding appearance—and by their psychometric properties—factor load, correlation with the total scale score—and clinical utility. The initial psychometric study of DAS-24 used a sample of 518 participants, of both sexes, between 18 and 70 years. The appearance issues in the sample were due to the trunk, sex organs, upper limbs, lower limbs, face, head, or neck. Multivariate psychometric analysis was not performed—exploratory or confirmatory factor analysis. However, in terms of reliability, in this study the DAS-24 showed good temporal stability in the re-test with an interval of six months (r = 0.82). In terms of evidence of validity, satisfactory evidence of concurrent validity was generated with DAS-59 (r = 0.88), discriminant validity with the measurement of positive affect (r = −0.24). Finally, evidence of convergent validity was also generated with measures of anxiety (r = 0.50), depression (r = 0.45), social avoidance (r = 0.53), fear of negative evaluation (r = 0.50), negative affect (r = 0.50), and shame (r = 0.66) [27].

A second psychometric study, with multivariate methods of analysis, was conducted later in United Kingdom [30]. The total of 1265 participants—614 in the community and 651 in hospital treatment—of both sexes aged between 18 and 91 years participated in this new study. Exploratory factor analysis was performed for a subsample of 500 participants, while using axis factoring extraction and oblique rotation. The exploratory factor analysis generated a bi-factorial structure, with factor 1, with 18 items, eingenvalue of 9.73–9.98, and factor 2 with six items, eingenvalue 1.85–1.91, the others being below 1, both in clinical and non-clinical samples, respectively [30]. Confirmatory factor analysis was then performed with (1) the rest of the sample; (2) only the clinical sample; (3) only the non-clinical sample; and, (4) a pool of 500 participants that were randomly selected from the entire sample. In the four cases, the bi-factorial model was confirmed. The mean of the Root Mean Square Error of Approximation (RMSEA) value was 0.06; Goodness of fit index (GFI) = 0.86; Adjusted Goodness of Fit Index (AGFI) = 0.83; Normed Fit index (NFI) = 0.84; Non-Normed of Fit Index (NNFI) = 0.86; and, Comparative Fit Index (CFI) = 0.88. Cronbach’s alpha values ranged from 0.93 to 0.79 between the factors and samples [30].

DAS-24 has already been validated in Thailand [31] and Portugal [32]. In Thailand [31], the psychometric study used a sample of 208 participants, with visible changes in appearance, of both sexes, aged 19−80 years. The preliminary analysis of the data resulted in the elimination of items 14, 17, and 22 because of their asymmetry, and items 1 and 8 for their low correlation with the total score of the scale. Exploratory factor analysis was then performed, using main component analysis with oblique and varimax rotations as extraction. The factor solution of the DAS-24 Taiwanese version, now with 19 items, has three factors—social distress, social avoidance, and negative affect—generated by the main components analysis with oblique rotation. Cronbach’s alpha value was only calculated for the instrument as a whole—unlike the recommendations [33]—and a value of 0.91 was found.

The validation study of DAS-24 in Portugal [32] was conducted with a sample of 508 participants with no specific clinical condition, between 17 and 89 years old, of both sexes. Exploratory factor analysis was performed with main components analysis with varimax rotation, forced for one factor in order to verify the correlation pattern between the items. Items 8, 15, 20, and 24 presented weak (0.50) and moderate (>0.50 to 0.80) correlations. The confirmatory factor analysis, performed with the same sample used in the previous analysis—unlike the recommendations [33], presented an initial adjustment that was tolerable—χ2/*df* = 4.7; RMSEA = 0.09; GFI = 0.80; CFI = 0.79; NFI = 0.77 for the one-factorial model. The adjustment of the model was due to the acceptance of several communalities, which enabled the preservation of all items. However, it should be noted that some items had very low factorial load, such as item 15 (λ = 0.15), item 8 (λ = 0.21), item 20 (λ = 0.21), and item 24 (λ = 0.33). In addition, the acceptance of error covariance foresees theoretical reason that is linked to the mathematical advantages to fit the model. What can be observed in the authors’ decisions was leniency in this criterion—for example, the acceptance of the “feeling rejected” item 14 and item 18 “worrying about not wearing the favorite clothes. With these modifications, satisfactory adjustment was achieved, χ2/*df* = 2.5; RMSEA = 0.05; GFI = 0.91; CFI = 0.93; NFI = 0.91.

Our literature review indicated that there was a cross culturally adapted version of DAS-24 into Brazilian Portuguese, but no psychometric assessment was carried out [34]. The psychometric validation of this scale for PLHA will make it possible to know the nature, extent, and determinants of individual differences in disease impact and diagnosis. It may also allow different health professionals to monitor the impact of their interventions on the affective dimension of body image in PLHA, adjusting and adapting them in order to make them a resource for the restructuring of body identity.

Recognizing the importance of having an outcome measure to allow for tracking changes of body image during treatment and for having the methodological gap of a valid instrument and appropriate to obtain such a measure, the aim of this research is to assess the psychometric properties of the Brazilian Portuguese version of Derriford Appearance Scale 24 (DAS-24) with a reference sample of PLHA.

## 2. Materials and Methods 

### 2.1. Study Design and Sample Size Calculation

This is a methodological study with a non-probabilistic sample [35] having, as target population, Brazilian people living with HIV/AIDS. The sample size was calculated according to the recommendations of Hair et al. (2009), which recommend that the sample size should be five participants per assessed parameters. When considering that the largest model of DAS-24 tested has 51 parameters (24 observable variables; 24 errors; three first-order latent variables), the minimum sample size for this research should be 255 participants.

### 2.2. Participants

The sample consisted of 400 participants (male = 67.8%, *n* = 271), who met the following inclusion criteria: being over 18 years old, reactive to HIV and on drug treatment. The participants’ age ranged from 18 to 78 years (M = 41.12 ± 11.68). Of the total, 35% (*n* = 141) reported female identity and 64.8% (*n* = 259) reported male identity. Regarding housing, 39.8% lived with their spouse (*n* = 159), 22.5% lived alone (*n* = 90), and 37.8% lived with family members (*n* = 151). Of the total, 53.8% (*n* = 215) declared themselves caucasian, 9.3% (*n* = 37) black, and 43% (*n* = 148) of other ethnicities.

The participants that were included in the study met the following criteria: being over 18 years of age, being a HIV seroreagent individuals, and being under regular treatment for HIV/Aids. Comorbidities were not screened. Patients returning from treatment after a period longer than six months of drop-out were excluded.

### 2.3. Instruments

Brazilian Portuguese version of Derriford Appearance Scale 24 (DAS-24) for people living with HIV/AIDS in Brazil [34]. DAS-24 is originally from the United Kingdom and it was created in order to evaluate the distress and difficulties of those who have changes in appearance [27]. Items 1, 4, 8, 10, 11, 14, 17, 20, and 22 are arranged on a four-point Likert scale (1 = not at all, 4 = extremely; 1 = never, 4 = always) as are items 2, 3, 5, 6, 7, 9, 12,13, 15, 16, 18, 19, 21, 23, and 24 on the which only adds the “not applicable” option (with zero score). There are two additional items, which are not included in the structural model or in the sum of the score, but that provide information regarding the intensity of pain and the frequency of physical limitations that the respondent may have due to the change in their appearance. The higher the mean score, the greater the anguish and concern for appearance. [27]. The Brazilian Portuguese version of DAS-24 was made according to the guidelines that were recommended by Beaton, Bombardier, Guillemin, and Ferraz (2002) [36], having PLHA individuals on their sampling.

Demographic Questionnaire: specially designed for this research and only consists of the identification of biological sex, gender, age, marital status, and ethnicity.

### 2.4. Procedures and Ethical Aspects

This study was approved by the Research Ethics Committee of the ABC School of Medicine and it is registered under number: 092257/2014. Patients from the HIV/AIDS ambulatory of São Bernardo do Campo were invited to participate in the research in an individualized approach, by the professionals of the nursing team with which they were familiar and felt confidence. Only those who agreed to participate voluntarily were given the informed consent form (ICF). The questionnaires were answered by the participants, in the outpatient clinic, in a private space. The participants individually completed the instruments, taking between 15–25 min. to complete.

### 2.5. Investigated Models

In this research, model fit was investigated in three structural models, previously established for the DAS-24. The first model is the one-factor solution established the study of the development of the instrument [27] and ratified in the Portuguese version of the instrument (Mendes et al., 2016). In this model, low factor loadings for items 8, 15, 20, and 24 were noted.

The second model investigated was a two-factor solution, which achieve good fit in a clinical and non-clinical sample [30]. The factors in this model are: (1) general self -consciousness of appearance (GSC), items # 1, 2, 3, 4, 5, 7, 8, 9, 10, 11, 13, 14, 17, 19, 20, 22, 23, and 24, and (2) Sexual and body self-consciousness of appearance (SBSC), items # 6, 12, 15, 16, 18, and 21.

The third model tested was a three-factor solution, which was created for a clinical and non-clinical sample in Thailand [31]. The factors of this model are; (1) social distress (SD), items # 6, 9, 10, 11, 13, 16, 18, 19, and 23; (2) social avoidance (SA), items # 5, 12, 15, 21, and 24; and, (3) negative affect (NA), items # 2, 3, 4, 7, and 20 (Figure 1).

### 2.6. Statistical Analysis

The PRELIS^TM^2 module of the LISREL^®^ system was used to prepare the data for the Confirmatory Factor Analysis (CFA), adopting the listwise deletion criterion for missing data [37]. After generating the PRELIS^TM^2 file, the SIMPLIS model, which is a coding of the LISREL^®^ system, was used. The CFA was then generated, enabling the estimation of the parameters of the construct measurement model. Given that the data distribution was not in line with the normal multivariate distribution, the Unweighted Least Square extraction method was used, because it was not sensitive to this violation [38].

The following adjustment indexes were also considered: goodness-of-fit index (GFI), adjusted goodness-of-fit index (AGFI), normed fit index (NFI), non-normed fit index (NNFI), and comparative fit index (CFI). The Root Mean Square Error of Approximation (RMSEA) was also considered. The model was accepted when χ2/*df* ≤ 5; RMSEA ≤ 0.08; CFI, NFI, NNFI, GFI, and AGFI ≥ 0.95 [33]. For models with unsatisfactory initial adjustment, factor loadings, residuals, and LISREL^®^ system modification indexes were considered for adjustment.

In the analysis of the measurement model, the construct validity (discriminant and convergent) and internal consistency were considered. In order to establish the internal consistency, Cronbach’s alpha (α) and construct reliability (CR) were calculated and values that were above 0.70 were considered to be acceptable [33,39]. The CR is given by the formula: (square of the sum of the factor loadings)/(square of the sum of the factor loadings) + (sum of the errors of the observable indicators).

In order to establish convergent validity, factor loadings and the Average Variance Extracted (AVE) were considered. The AVE is given by the formula: (sum of the square of the factor loads)/(sum of the square of the factor loads) + (sum of the errors of the observable indicators). Values that are above 0.50 were accepted (Hair et al., 2009) and factor loadings (λ) above 0.50 are preferable and above 0.40 are acceptable (Hair et al., 2009).

Discriminant validity was assessed by comparing the shared variance with the AVE of each latent variable. Evidence of discriminant validity is established when a AVE is ≥0.50 and greater than the shared variance between factors [40].

Finally, correlation analyses were performed between the DAS-24 scores and the participants’ age, and differences in the DAS-24 score in relation to gender and gender role were analyzed, in order to generate some additional information. The part of the body that bothered them, the type of lipodystrophy, and the other body feature that bothered the participants other than the one first described. The discrete variables of this analysis (age and DAS-24 score) were adherent to the normal distribution; hence, Pearson’s correlation test and Student’s t-test were used. For all tests, a 95% confidence interval was considered. SPSS 21 and LISREL^®^ software were used for these analyzes.

## 3. Results

### 3.1. Factorial Structure

The first estimation of model one showed adequate adjustment indices, except for RMSEA, χ^2^/*df* = 4.37, RMSEA = 0.092, CFI = 0.99, NFI = 0.98, NNFI = 0.99, GFI = 0.98, and AGFI = 0.98. To better fit the model, items 8 and 20 were removed for their low factor loadings, λ = 0.15 and λ = 0.36, respectively. Item 17 was eliminated by various and high errors that are associated with it in the model. In the inspection of LISREL^®^ modification indexes, the covariance of the errors between items 4 and 10, 9 and 23, 11 and 14, and 14 and 22 had conceptual coherence and, therefore, were accepted in the model. Each modification occurred with each new estimation. The final model achieved satisfactory fit in all parameters, χ^2^/*df*l = 3.46, RMSEA = 0.079, CFI = 1, NFI = 0.99, NNFI = 0.99, GFI = 0.99, and AGFI = 0.99 (Figure 2A, Table 1).

On first estimation of model two, once again, only the RMSEA index was below the reference value, χ^2^/*df* = 4.21, RMSEA = 0.090, CFI = 0.99, NFI = 0.98, NNFI = 0.99, GFI = 0.98, AGFI = 0.98, and items 8 (λ = 0.15) and 20 (λ = 0.36) had low factor loads, which were eliminated in this order in two new rotations. The other adjustments in the model are the same as those made in the adjustment of model one (the elimination of item 17 and the acceptance of covariance errors), which resulted in a model with satisfactory adjustment, χ^2^/*df* = 3.39, RMSEA = 0.078, CFI = 1, NFI = 0.99, NNFI = 1, GFI = 0.99, and AGFI = 0.99 (Figure 2B, Table 1).

Model three had a satisfactory adjustment in the first rotation, χ^2^/*df* = 3.69, RMSEA = 0.082, CFI = 1, NFI = 0.99, NNFI = 1, GFI = 0.99, and AGFI = 0.99. However, item 20 remained with low factor loading in this model (λ = 0.34), and its elimination only worsens the model fit in some parameters, χ^2^/*df* = 4.03, RMSEA = 0.087, CFI = 1, NFI = 0.99, NNFI = 1, GFI = 0.99, and AGFI = 0.99. The LISREL^®^ modification index did not indicate covariance errors that could be accepted, and the decision was to consider the first rotation of model 3 as final (Figure 2C, Table 1).

When considering all of the models after their fit, model two had the best structural fit, and model three the worse. However, for model two, the covariance of its factors was strong, having no statistical evidence that they are really distinct of each other regarding then (AVE < r^2^). Thus, the decision is to adopt the model one as a structural model for the Brazilian Portuguese version of DAS-24.

### 3.2. Converged Validity and Internal Reliability of Model One

Regarding the convergent validity, when considering the AVE, it was below the acceptance limits (AVE = 0.46). However, all of the items were above the acceptable value, ranging from λ = 0.40–0.81. The construct reliability (CR = 0.94) and Cronbach’s alpha (α = 0.94) values were above the reference value (α and CR ≥ 0.70). Discriminate validity was not evaluated, since Fornel and Larcker’s criteria does not apply for this model

### 3.3. Extra Items

Through Pearson’s correlation test, it was possible to verify that the two extra items (“The characteristic of my appearance that bothers me causes me physical pain/discomfort” and “The characteristic of my appearance that bothers me causes physical limitations to do what desire”) have a high positive association with the scores of the Brazilian version of DAS-24, r = 0.61 (*p* < 0.05) and 0.68 (*p* < 0.05), respectively.

### 3.4. Additional Analyzes

There is significant difference, t (398) = −5.83, *p* < 0.001, *d* = 0.64, between the male (*n* = 271, M = 27.23 ± 14.86) and female (*n* = 129, M = 37.53 ± 17.24) participants, in the score of the Brazilian Portuguese version of DAS-24, with higher scores in females. There are also significant differences in DAS-24 scores, t (397) = −6.27, *p* < 0.001, *d* = 0.67, in relation to gender identity, women (*n* = 140, M = 37.42 ± 16.89) had higher scores than men (*n* = 259, M = 26.77 ± 14.84). 

There is a significant difference, t (398) = 15.98, *p* < 0.001, *d* = 1.61, between those participants who have some characteristic that bothers them (*n* = 207, M = 40.43 ± 14.54) and those who do not (*n* = 193, M = 19.96 ± 10.61), with the latter having the lowest score. Regarding the body characteristic that makes the participant feel most concerned or bothered, there is no significant difference between upper limbs (*n* = 3), lower limbs (*n* = 8), abdomen (*n* = 41), buttock (*n* = 5), face (*n* = 62), and other body parts (*n* = 88), F (5201) = 0.32, *p* = 0.90. Likewise, there is no significant difference between lipodystrophy types and other appearance issues affecting participants, when considering facial lipoatrophy (*n* = 39), upper limb lipoatrophy (*n* = 3), lower limb lipoatrophy (*n* = 8), buttocks lipoatrophy (*n* = 5), abdominal lipohypertrophy (*n* = 43), and other changes in appearance (*n* = 107), *F*(6, 194) = 0.44, *p* = 0.85. When there was any other characteristic (other than that described as the “characteristic of my body that bothers me”) that did not please the participant, there was no significant difference either, *F*(6, 89) = 0.95, *p* = 0.46, when considering facial lipoatrophy (*n* = 7), upper limb lipoatrophy (*n* = 2), lower limb lipoatrophy (*n* = 5), buttocks lipoatrophy (*n* = 8), abdominal lipohypertrophy (*n* = 14), other characteristics (*n* = 38), or the combination of more than one characteristic (*n* = 22). Finally, there is no significant correlation between the Brazilian Portuguese version of DAS-24 score and age, r = −0.001, *p* = 0.98.

## 4. Discussion

The aim of this paper was to assess the psychometric properties of the translated and culturally adapted version of Derriford Appearance Scale, with 24 items (DAS-24) [10,34], for a Brazilian sample of PLHA.

Of the three models tested, the one that concomitantly obtained the best structural adjustment and best fit of the measurement model was the one-factor model, after eliminating items 8, 17, and 20 and accepting the covariance of errors between items 4 and 10; 9 and 23; 11 and 14; and, 14 and 22. It is worth mentioning that the changes made do not imply the loss of the quality of the measure. The elimination of items in cross-cultural methodological studies is not uncommon, including attitudinal measures in health [41]. Specifically, for DAS-24, items 8 and 20 were also eliminated in the methodological study that was conducted in Portugal [32]. 

It is unlikely to have occurred a bias on the items during the cross-cultural adaptation process that may have caused this elimination, since the work was thoroughly conducted and accompanied by the original author of the instrument, as reported by Martins et al. (2019). It is also noteworthy that the cross-cultural adaptation study for Brazil also pointed out that item 8 was potentially fragile, due to its low correlation with the total score of the instrument [34], a fact that does not make its elimination strange. It is more plausible that cultural differences have led to this situation: specifically, it is possible that there are some aspects of PLHA body experiences that are specific to Brazil and that may have an impact on how they deal with changes in appearance, due to the chronicity of the disease and virus activity. From this perspective, the eliminated items could be considered as less relevant to the construct under study, or rather could be observable variables that are unexplained for the latent variable. Indeed, previous studies have suggested discrepancies between scales for assessing body image developed in the Western and translated into Brazilian Portuguese [42].

As for the covariances of errors that are accepted in this model, common causes include item redundancy (caused by similar content or social desire) and/or omission of an exogenous factor [43]. The acceptance of the covariance error should be theoretically supported rather than a purely statistical reason, for example, to improve model fit [38,44]. Having this clear criterion helps to avoid creating model caricatures and ignore the logic of confirmatory factor analysis. Social desirability is unlikely to be a factor for the covariance of error, as the conditions for data collection to reduce this bias (voluntary and anonymous participation) were guaranteed. It is also unlikely that an ignored latent variable will be absent in DAS-24, since the analyzed models have already been proposed and investigated by previous studies and the items of the scale already thoroughly investigated. It was the content analysis of the paired items that led us to consider that the existence of error covariance was due to a similar content between items.

Weak evidence of convergent validity of the measurement model was generated, but still relevant. Although the average variance extracted was within the citation limit, the factor loadings were adequate, which indicated that the items inserted in the single factor were coherent with each other. It was also observed that the additional analyzes made a point to discriminant validity of the measure, since they indicated differences between biological sex and gender identity, as predicted by the body image’s literature [45,46]. The significant difference between the scores of participants who have some feature that bothers them and the one who does not was also relevant evidence, since it demonstrates that the measure can distinguish distress and beliefs caused by these changes in appearance, which becomes the theoretical essence of this measure [27,28].

The non-statistical significance of the impacts of different appearance changes leads us to resume the stigmatization that is generated by the signs of chronicity of the action of the HIV virus and disease [14]. What our evidence complains is that it does not matter what the change is (whether on the face, trunk, limbs, back, or other body location), but that it may be associated with HIV/AIDS. Finally, it is commented that the non-association of the score of the Brazilian Portuguese version of DAS-24 with age leads us to consider that distress and beliefs that are generated by appearance alteration are independent of age; thus, both young and old. Older people need to receive the same degree of support from the care service in order to promote the positive resignification of HIV/AIDS.

The present study validated DAS-24 for a specific clinical group of Brazilians: people living with HIV/AIDS. Despite the evidence psychometric evidence generated here, a number of limitations must be considered. The opportunistic method of participant recruitment means that the current results should only be considered for the reference population of the instrument. On the other hand, this study presents contributions to the improvement of DAS-24 by the possibility of further cross-cultural studies. Because appearance is considered to be an important part of the Brazilian body image [47], it would be interesting to have DAS-24 validated for other groups in which appearance changes, such as burns, vitiligo, psoriasis, with progressive hemifacial atrophy (Parry–Romberg syndrome), among others. As recruitment was done on an outpatient basis with limited data collection time, future studies could also examine the extent of relationships and associations of body image impact assessed by DAS-24 with other relevant constructs, such as body dysmorphia, acceptance of plastic surgery, social physical anxiety, quality of life, and physical attractiveness in the Brazilian context.

## 5. Conclusions

A satisfactory fit was achieved for the original model with the reference sample of PLHA, with weak evidence of construct validity and evidence of internal consistency for the Brazilian Portuguese version of DAS-24.

## Figures and Tables

**Figure 1 healthcare-08-00569-f001:**
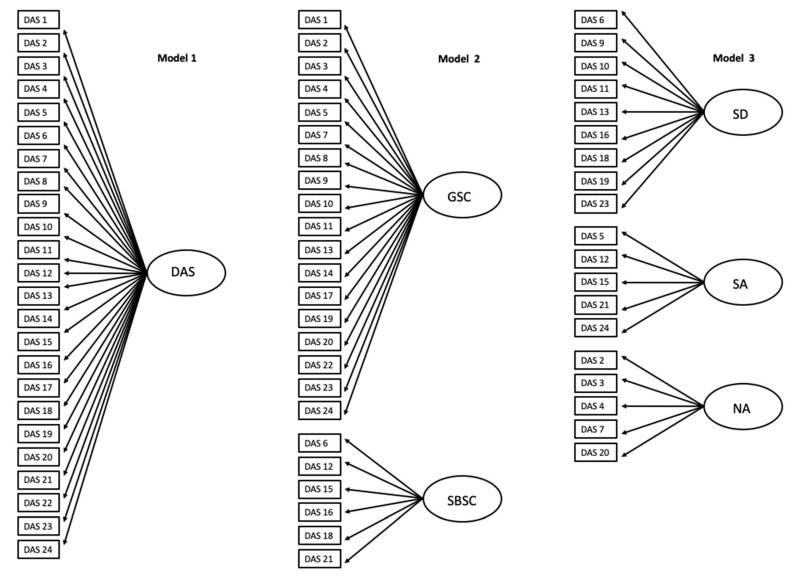
Structural models investigated from the Brazilian version of *Derriford Appearance Scale 24* (DAS-24). Note: GSC: General Self-Consciousness of Appearance; SBSC: Sexual and Body Self-Consciousness of Appearance; SD: Social Distress; SA: Social Avoidance; NA: Negative Affect.

**Figure 2 healthcare-08-00569-f002:**
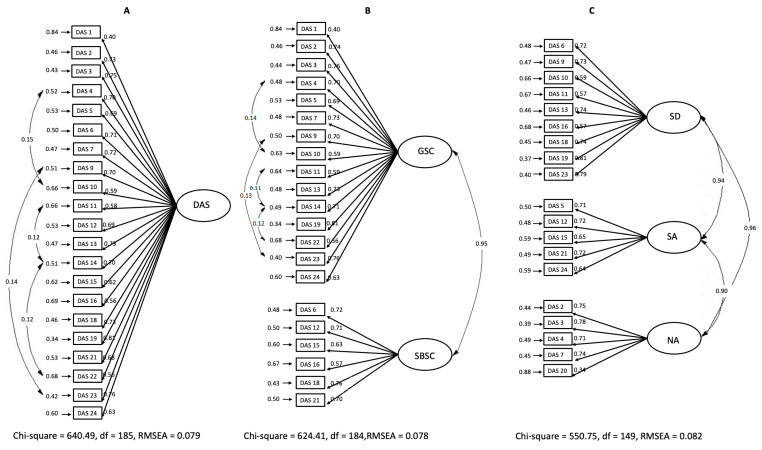
Adjusted model for Brazilian Version of DAS-24. (**A**) unifactorial Model; (**B**) bifactorial model and (**C**) three-factorial model

**Table 1 healthcare-08-00569-t001:** Adjustment indexes of measurement and measurement model for DAS-24 in the three models.

Model	*n*	>λ	< λ	χ^2^/*df*	CFI	NFI	GFI	RMSEA	AIC	α	CC	AVE	R^2^
1—Inicial	400	0.15	0.80	4.37	0.99	0.98	0.99	0.092	1197.71	0.93	0.93	0.41	-
1—Ajusted		0.40	0.81	3.46	1	0.99	0.99	0.079	732.49	0.94	0.94	0.46	-
2—Inicial	400	0.15	0.80	4.21	0.99	0.98	0.98	0.090	1156.61	0.84–0.91	0.92–0.84	0.42–0.47	0.88
2—Ajusted		0.40	0.81	3.39	1	0.99	0.99	0.078	719.41	0.84–0.93	0.92–0.84	0.46–0.47	0.90
3—Inicial *	400	0.32	0.81	3.69	1	0.99	0.99	0.082	632.54	0.79–0.89	0.81–0.89	0.47–0.49	0.88–0.92

Note: >λ = lower factor loading, >λ = higher factor loading, χ2/df = standard chi-square, CFI = Comparative fit index, NFI = Normed fit Index, RMSEA = root mean square error of approximation, α = Cronbach’s alpha; CR = construct reliability; AVE = average variance extracted; R2 = shared variance. Inicial* = final.

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
