# Peer review of "Psychometric Validation of the Brazilian Portuguese Version of the Derriford Appearance Scale-24 (DAS-24) for People Living with HIV/AIDS"

_healthcare, 2020, doi:10.3390/healthcare8040569_

Round 1
Reviewer 1 Report
The authors have tried to study the psychometric properties of a previously translated instrument. Despite the kind of this kind of study, it is necessary to add more information to increase its quality. However, the first thing that strikes me is the reason why they intend to publish two articles that analyse the same idea. That is, I think that when a research team translate a test, it is also important to assess its psychometric properties. Therefore, it would be necessary to answer this issue and to avoid a “salami slicing” manuscript.
I would like to know whether authors previously calculated the required sample size to conduct this study. In fact, it would be necessary to add in their manuscript.
It would be necessary to assess or to include in the manuscript whether the included sample is or not a good representation of this population (e.g. people infected with HIV).
I would like to know the reason why this questionnaire is not very widespread. In fact, cross-cultural validity could be a questionnaire.
I would like to know if the authors could include other previously validated questionnaires with which to compare the questionnaire. What is the reason that they have not used other questionnaires related to the subject?
I think there is an imbalance in the topics covered in the introduction. In fact, I believe that they have only devoted a paragraph to the analysis of the issues, but that this is insufficient. In this sense, I consider that the authors should treat the questionnaire in greater depth and include the psychometric properties of the original instrument. Furthermore, I consider that they should defend the usefulness of translating this, as well as whether it is a good instrument or not. Nor do I see that they have commented on the limitations of it.
I consider it important to know if the authors have controlled for the effect of sociodemographic variables, since they are not evaluating a homogeneous population.
Finally, I have detected some typographical errors in the abstract (for missing data. he adjustment of three) and in the introduction (lines 93-94).
Author Response
Revisor 1
1 - The authors have tried to study the psychometric properties of a previously translated instrument. Despite the kind of this kind of study, it is necessary to add more information to increase its quality. However, the first thing that strikes me is the reason why they intend to publish two articles that analyse the same idea. That is, I think that when a research team translate a test, it is also important to assess its psychometric properties. Therefore, it would be necessary to answer this issue and to avoid a “salami slicing” manuscript.
Answer: You made an excellent point and we, authors, discussed this same concern prior of manuscripts submission. However, we took the decision to work the two parts of our research in two different scientific papers for some reason that I am going to present now.
(1) the process of culturally adapt a scale is time consuming and demand lots of work and qualitative analysis for conclusion. There are at least 20 different methodological protocols to cover all the steps needed for this work, and we particularly adopted the Beaton, Bombardier, Guillemin and Ferraz (2002) with Ferreira, Neves, Campana & Tavares (2014) adjustments. It includes 5 steps: (1) 2 independent translations; (2) a synthesis of the translations; (3) 2 Back translations of the synthesis; (4) a experts committee to analyze all material produced and evaluate idiomatic, semantic, cultural and conceptual equivalence of answer option, items and instructions, and (5) pre-test round to evaluate lay out adequacy, clarity and comprehensiveness of the items, instructions and answer options. To describe all this qualitative work we need space, and because the length of the articles, this part is usually neglected in papers in which a cross culturally process and the psychometric study is published all together.
(2) The authors had some experience in publishing this kind of work and we made the choice in the past to publish in this way: just briefly mentioning the cross-cultural process and emphasizing the psychometric work – which, of course is the most robust part of the work. However, we receive considerable number of emails requesting information about the cross-cultural work, by researches interested running the psychometric study in a reference sample different from our work. To do so, they need detail from the qualitative work made on the measure.
(3) Once made, the cross cultural adaptation is available for the target country - except in situations where there are very strong cultural niches, and then maybe a second study for this group should be done. What we see is the repletion of this work: we have scales, as Body esteem Scale – just to name one - with three different adaptation studies! It is research money and time spent to do a work which is already done – and can create confusion and difference on data collection across the versions. If the cross cultural study of the scale is a free access, detailed, easy to find we may prevent this kind of “reinventing the wheel” study. On the other hand, the psychometric study in unique and need to be done in the same culture, for different references samples. Scales with several psychometric studies are the best scenario, since the researcher interested to in using the instrument, may found a specific structure adjusted for his reference sample: the most adequate scenario.
(4) Despite this, we understand that bring a scale for a new target country for the first time and run a psychometric study is a two-phase study. The first part, the cross-cultural adaptation of the measure is clearly qualitative, asking for extensive and detailed steps description, with discussion regarding the theory that supports the measure. And it is a study primarily direct for a local audience - since the article will be sound by researchers of the target country in a first place. The other phase, the psychometric is clearly quantitative, with different aims and a specific method, requiring a discussion regarding the evidences and considerations of their lack or presence for the reference sample used, in the light of the theory that supports the measure and the psychometric theory as well. It usually grabs more attention in a international context, since it can provide data from a different culture and for researchers interested in multicentric data collection, announces that a specific measure is available in a new target country.
Considering in one hand, the experience of being inquired about the scale cross cultural process, to avoid the “re-done” work for cross- culturally adapt DAS-24, and the limitation of the length in journals for their articles we decides to assume that this was a two part study, in which the first one is clearly directed for the process of the cross cultural work and for a internal audience – that’s why a Brazilian Journal was chosen - and the second one, in directed to present the psychometric study, for the local and international audience, and that’s why we made of choice for Healthcare.
References:
Beaton, D. E., Bombardier, C., Guillemin, F., & Ferraz, M. B. (2002) Recommendations for the cross-cultural adaptation of healthy status measures. Chicago, IL: American Academy of Orthopedic Surgeons/Institute for Work & Health.
Ferreira, L., Neves, A. N., Campana, M. B., & Tavares, M. D. C. G. C. (2014). Guia da AAOS/IWH: sugestões para adaptação transcultural de escalas. Avaliaçao Psicologica: Interamerican Journal of Psychological Assessment, 13(3), 457-461.
2 - I would like to know whether authors previously calculated the required sample size to conduct this study. In fact, it would be necessary to add in their manuscript.
Answer: great observation. We also agree that it is an important. I kindly would like to ask you to read aging our subsection 2.1. Study design and sample size calculation, since we presented this information in the last sentence (former Lines 130-132; now 199-201). Psychometrics has three pocket rules for sample size calculation and we used the one that asks five participants for each parameter – being the parameters the variables, the variable paths and the associated errors, and presented by Hair, Anderson, Tathan and Black (2009)
3 - It would be necessary to assess or to include in the manuscript whether the included sample is or not a good representation of this population (e.g. people infected with HIV).
Answer: So sorry, but I must disagree of the importance of this point in a psychometric study. Consulting the psychometrics requirements for sample we were unable to find a strong argumentation of this need for one main reason: all the conclusions are regarding the measure, not the sample. So , in first place we need to be sure that the sample size is enough to run all the robust tests keeping their power. Just to be sure, we calculated the power of our sample in M-Plus (another program to run CFA) and it was .98. Reading the mains results, they address evidence of reliability and validity for the scale. The focus is not the persons with HIV, but the scale. That’s why a concern with representation – and because this samples are usually non-probabilistic and in essence non representative – is minor.
However, I might agree that something must be made, because we added some additional analysis – focusing in giving for the international audience some evidence regarding body image changes concern of PLHA in Brazil. Hence, some conclusion can be made for the sample we added the size effect (lines 338-343) for the significant results, so the reader is able to make the judgment in considering this evidence. Thank you for it.
4 - I would like to know the reason why this questionnaire is not very widespread. In fact, cross-cultural validity could be a questionnaire.
Answer: Well, we must say at first that DAS-59 e DAS-24 are the golden measure to evaluate body image changes in persons with great body appearance modifications – such in burns, illness, operations to remove a body part or a limb. Just to make this point, we would like to quote Cogliandro, Barone, Ciarroschi, Salsillo, Moss, Tamone & Persichetti (2020) in his systematic review study regarding these measures:
The literature review yielded 28 articles with DAS59 and 32 articles with DAS24 from peer-reviewed journals met our inclusion criteria and were included in the analysis. Since its inception in 2001, the number of publications incorporating the DAS has increased each year with a total of 3.483 patients for DAS59 and 6.012 patients for DAS24. The prospective study design was the most prevalent, being used in 50% of publications (n = 30), 57% for DAS59, 43% for DAS24. The DAS59 was administered in a cross-sectional design in 21% of studies (n = 6) and DAS24 41% (n = 13). The DAS59 was administered in a retrospective design in 18% of studies (n = 5) and DAS24 9% (n = 3). We reported only one case series for DAS59 and 2 for DAS24.
Regarding the cross-cultural study and psychometric evaluation of the measure the copyright of the measures may be a point to consider. When we asked Dr Moss his permission to do the study this discussed, since he firstly asked to keep the copyright of the Brazilian version. We argued that data would be collect in a public center of treatment and a return for their consent should be their free use of the scale. A professor from a public university and public works also were part of the research. So, in face of this points and because we invited him to do all the steps with us as a partner to keep the control of his measure we received his “bless”. In fact, Dr Moss worked with us from the beginning to the end. However, with a different scenario, this maybe not the case…and we may see a scarcity of cross cultural research with DAS-59 and 24 because of the copyright issue. That why our research has an extra value J
Reference:
Cogliandro, A., Barone, M., Ciarrocchi, S., Salzillo, R., Moss, T., Tambone, V., & Persichetti, P. (2020). A systematic review on the Derriford Appearance Scale (DAS) questionnaire in surgical research. European Journal of Plastic Surgery, 1-14.
5 - I would like to know if the authors could include other previously validated questionnaires with which to compare the questionnaire. What is the reason that they have not used other questionnaires related to the subject?
Answer: I will answer your observation in two parts. First, we do not have other body image scales in Brazil validated for this reference sample, as these works showed
Laus, M. F., Kakeshita, I. S., Costa, T. M. B., Ferreira, M. E. C., Fortes, L. D. S., & Almeida, S. S. (2014). Body image in Brazil: recent advances in the state of knowledge and methodological issues. Revista de Saúde Pública, 48, 331-346.
Carvalho, P. H. B. D., & Ferreira, M. E. C. (2014). Imagem corporal em homens: instrumentos avaliativos. Psicologia: Teoria e Pesquisa, 30(3), 277-285.
They are not exactly recent, but we are observing the evolution of the literature since our work started, and regarding our sample reference, nothing changed. Today, just to give an actual glance of the current status of body image measures in brazil we made a search on google schoolar - https://scholar.google.com/scholar?start=0&q=%22body+image+scales%22+brazil&hl=pt-BR&as_sdt=0,5 – and nothing changed, and no scale are properly available to collect data and run correlation tests.
Second, we chose a psychometric approach that uses multidimensional analysis and robust tests to evaluate convergent and discriminant validity, and internal consistency. We understand that we work with a psychometric protocol different from that recommended by the American Educational Research Association (AERA), American Psychological Association (APA), & National Council on Measurement in Education (NCME), & Joint Committee on Standards for Educational and Psychological Testing - the “Standards for educational and psychological testing” – which uses additional measures to create evidence of validity. The point we would like to make here is that, although distinct, the ours is adequate.
In this work and in our research we rely on the tripartite model of Nunally & Bernstein (1978) also followed by Anastasi and Urbina (2000), Marôco (2014) and Hair, Anderson, Tatham and Black (2005). From this perspective, validity is “the ability of the indicators of a construct to accurately measure the concept under study” (Hair et al., 2005, p.470). The theoretical framework of these authors established the foundations of psychometric assessment and theory, continuing, in our current opinion, useful and easy to apply for essentially two reasons:
- i) the easy calculation of convergent and discriminant validities based on confirmatory factor analysis (proposal by Fornell & Larcker, 1981). Here, convergent validity is established when the variance related to the measurement error is less than the captured variance. This is established through the extracted average variance (AVE), which must be greater than 0.50, using the formula:
In other words, the AVE> 0.50 indicates that the factor loads of the observable variables (items) converge satisfactorily to represent the latent variable (factor) as theoretically predicted. The discriminating variable is established when the shared variance (r2) between the factors is less than the stroke ... this shows us that the "similarity" between the factors is inferior to its internal "logic", pointing out that the factors can be in fact discriminated against, since it is established that, despite covariates, the factors are inherent dimensions in the data.
In the specific area of ​​body image, this approach is recognized and recommended as a way to assess convergent and discriminant validity (Swami & Barron, 2018)
In addition, we understand the validity of the discriminating criterion as an estimate that goes beyond the existence of high / low or positive / negative correlations, but we understand this estimate as the ability of the measure to be discriminated between groups.
ii.) the need for no other “convergent” / “divergent” measures as required by the method of Campbel and Fisk (1959) to interpret the validity of the instrument in the sample, since these measures do not always exist or are easily identified. This is especially relevant for the present study, since we are not aware of the availability of other body image measures validated in Brazil for people living with HIV / AIDS.
For these reasons, we chose to use confirmatory factor analysis to assess the validity of psychometric instruments following the theoretical model presented by Anastasi and Urbina (1988). Without underestimating the usefulness of the additional complexity of the psychometric assessment proposed by the Standards of AERA, APA and NCME, we understand that the psychometric protocol used is a relatively expeditious assessment that the researcher can use to attest to the initial psychometric qualities of his instruments and data without the need for more complex analyzes that require additional measures that are not always easy to obtain (eg.: an external criterion) or do not even exist. On the other hand, the information necessary for the assessment of construct validity and reliability is easily accessible in a confirmatory factor analysis.
References:
Anastasi, A. & Urbina, S. (2000). Testagem psicológica. Porto Alegre: Artes Médicas.
Campbell, D. T., & Fiske, D. W. (1959). Convergent and discriminant validation by the multitrait-multimethod matrix. Psychological bulletin, 56(2), 81
Fornell, C. & Larcker, D. (1981). Structural equation models with unobservable variables and measurement error. Journal of Marketing Research, 18, 39-50.
Hair, J. F., Black, W. C., Babin, B., Anderson, R. E., & Tatham, R. L. (2005). Multivariate data analysis (6th ed.): Prentice Hall.
Marôco, J. (2014). Análise de equações estruturais (2ª ed.). Lisboa: ReportNumber.
Nunally, J. C., & Bernstein, I. H. (1978). Psychometric theory.
Swami, V., & Barron, D. (2018). Translation and validation of body image instruments: Challenges, good practice guidelines, and reporting recommendations for test adaptation. Body Image, advanced online publication.
6 - I think there is an imbalance in the topics covered in the introduction. In fact, I believe that they have only devoted a paragraph to the analysis of the issues, but that this is insufficient. In this sense, I consider that the authors should treat the questionnaire in greater depth and include the psychometric properties of the original instrument
Answer: we cannot agree more. We were a little bit concern with the length of the manuscript. But we agree with your point and added a subsection on the introduction regarding DAS-24 previous psychometric studies (Lines 109-181)
7 - Furthermore, I consider that they should defend the usefulness of translating this, as well as whether it is a good instrument or not. Nor do I see that they have commented on the limitations of it.
Answer: We understand your concern. The consideration regarding the cross cultural adaptation of the scale is a topic covered elsewhere – as we explained in our first answers of your comments. The last paragraph of our discussion covered our considerations regarding the limitations of the study. We add an additional sentence prior to our consideration, toe make it more clear (lines 410 - 411).
8 - I consider it important to know if the authors have controlled for the effect of sociodemographic variables, since they are not evaluating a homogeneous population.
Answer: No, because this was not running as a stepwise regression analysis, but CFA runs a series of simultaneous dependence relations. To generate acceptable factor loadings – how the observable variable contributes for the explanation of the latent variable - we need some diversity at the reference sample, in this case, people living with HIV – to generate diversity on answers (if we work with an homogeneous sample and all answer “disagree” for an item…I do not have what to measure: it is true, does not need to be assess.). That is why we collect data from men and wone, from young and older persons, with different marital status and different financial security perceptions. On the other hand, despite the fact that this diversity ”helps” the study the scale is adjusted to be used in the future for the sample of reference sample, considering a wide range of its diversity.
9 - Finally, I have detected some typographical errors in the abstract (for missing data. he adjustment of three) and in the introduction (lines 93-94).
Answer: We REALLY do thank you for this unpleasant work and for all your comments. They made us sit and reflect about our positions and choices, and we think this is the win situation for a peer’s review process. Thank you.
Reviewer 2 Report
Introduction
The manuscript presents the DAS 24 validation, but in the Introduction it is worthwhile to outline other methods for body image examination e. g. MBRSQ Casha (the authors refer to this author many times in the text, but do not refer to the body image tests constructed by Casha?). It is worth noting other tools and commenting on the DAS 24 value in the context of other body image methods.
Material and Methods The group of respondents is 60% men, so can this affect the results of the study? It's worth commenting on :Could it matter to the gender? Age of respondents: 18 to 78 years old - this is a very large range. Description of the group of respondents :no clear factors of inclusion and exclusion from the study No description of the group in terms of other mental disorders, body disabilities (were they excluded from the study?)
Author Response
1 - The manuscript presents the DAS 24 validation, but in the Introduction it is worthwhile to outline other methods for body image examination e. g. MBRSQ Casha (the authors refer to this author many times in the text, but do not refer to the body image tests constructed by Casha?). It is worth noting other tools and commenting on the DAS 24 value in the context of other body image methods.
Answer: Indeed, Dr Cash is one of the most prestigious researchers on body image field, as you can see here:
Thompson, J. K., & Schaefer, L. M. (2019). Thomas F. Cash: A multidimensional innovator in the measurement of body image; Some lessons learned and some lessons for the future of the field. Body image, 31, 198-203.
Rodgers, R. F., Campagna, J., & Attawala, R. (2019). Stereotypes of physical attractiveness and social influences: The heritage and vision of Dr. Thomas Cash. Body image, 31, 273-279.
He is now retired – left Body Image Journal in 2018 – but his legacy, in terms of theory and measurement is undeniable. However, among all his research themes, Dr Cash did not work specifically - in terms of measures – with people with great body appearance changes - such in burns and illness. So, despite the fact that we created a considerable number of scales - ABCD: Assessment of Body-Image Cognitive Distortions; MBSRQ: Multidimensional Body Self-Relations Questionnaire; BIQ: Body-Image Ideals Questionnaire; SIBID Situational Inventory of Body-Image Dysphoria; BIDQ Body Image Disturbance Questionnaire; BIQLI: Body Image Quality of Life Inventory; BISS: Body Image States Scale; ASI-R: Appearance Schemas Inventory-Revised; BICSI: Body Image Coping Strategies Inventory; BESAQ: Body Exposure during Sexual Activities Questionnaire – none target the phenomenon of body image adaptation under a great body appearance change. DAS-59 was the first of doing so, and them DAS-24 was developed to make the same, in a less time consuming way. They both are the golden measures for this. So, we avoid to compare DAS to other measures to skip the orange and apples situation. However, in order to be clear and to give more information regarding the measure, we added a subtopic to cover DAS-24 previous psychometric studies (Lines 109-181)
2 - Material and Methods The group of respondents is 60% men, so can this affect the results of the study? It's worth commenting on: Could it matter to the gender?
Age of respondents: 18 to 78 years old - this is a very large range.
Answer: I understand you concern and if our target was to make conclusions about the sample – about the body image of our sample 0 and not about the measure – the validity and reliability evidence for he measure for the reference sample – I could not agree more. However, the study needs this diversity in the sample. I will explain myself: . To generate acceptable factor loadings – how the observable variable contributes for the explanation of the latent variable - we need some diversity at the reference sample, in this case, people living with HIV – to generate diversity on answers (if we work with an homogeneous sample and all answer “disagree” for an item…I do not have what to measure: it is true, does not need to be assess.). That is why we collect data from men and wone, from young and older persons, with different marital status and different financial security perceptions. On the other hand, despite the fact that this diversity ”helps” the study the scale is adjusted to be used in the future for the sample of reference sample, considering a wide range of its diversity.
3 - Description of the group of respondents: no clear factors of inclusion and exclusion from the study
Answer: So sorry, we added this information now (Lines 209-212).
4 - No description of the group in terms of other mental disorders, body disabilities (were they excluded from the study?)
Answer: That was not a concern. Our sample were composed by patients of a public reference sample of HIV treatment and we invited them to answer the scales considering our inclusion criteria. Thank you for all your comments
Round 2
Reviewer 1 Report
Even though authors have tried to answer all my comments, I still have certain concerns that I hope the authors will solve for future studies.
Although the authors have defended the need to publish the same or very similar article in two parts, I consider that the arguments they have provided do not defend this replication. In fact, the need for space can often be solved by providing tables or supplementary information. A careful reading of both articles reveals that the information provided by one and the other is not enough to separate it into two parts.
The arguments provided by the authors regarding the population they have used to assess the questionnaire seem correct to me, but I continue to maintain that it is necessary to include the target population to which the questionnaire will be directed. In fact, it would add more value to the item right from the start.
Lastly, I thank you for taking the time to answer all the questions. I encourage you to continue working in this much-needed area and to continue your research progress in this area.
Author Response
Dear editor,
We answered the two additional points raised by our colleague. The change made to answer his second question is highlighted in yellow. We do thank you and ou reviewers for all consideration that our manuscript received and we hope we achieved the Healthcare standards for publication.
Reviewer 1
Q1.: Even though authors have tried to answer all my comments, I still have certain concerns that I hope the authors will solve for future studies.
Although the authors have defended the need to publish the same or very similar article in two parts, I consider that the arguments they have provided do not defend this replication. In fact, the need for space can often be solved by providing tables or supplementary information. A careful reading of both articles reveals that the information provided by one and the other is not enough to separate it into two parts.
Answer: Dear colleague, I understand your concern, but I still respectfully sustain the premise that the articles are inherently different, despite the fact that both works with the same measure. Methods and aims are completely different. The article with the cross-cultural work can be seen here: http://pepsic.bvsalud.org/scielo.php?script=sci_arttext&pid=S0104-12822019000200009.
This choice for publishing the cross-cultural process and the validity process in different papers – to be able to give the details and to do not lose the methodological focus on the articles – is not only ours. Just to illustrate this point of view:
Ex1: Tinnitus handicap inventory:
https://www.scielo.br/scielo.php?pid=S0104-56872005000300004&script=sci_arttext (cross cultural study)
and https://www.sciencedirect.com/science/article/pii/S180886941531048X (psychometric study)
Ex2: Nursing work índex – revised: http://repositorio.unicamp.br/bitstream/REPOSIP/77593/1/WOS000269670400007_I.pdf (cross cultural study)
https://onlinelibrary.wiley.com/doi/full/10.1111/j.1365-2702.2011.03776.x (psychometric study)
Ex3: Child Perceptions Questionnaire 11–14:
https://hqlo.biomedcentral.com/articles/10.1186/1477-7525-6-2 (cross cultural study)
https://link.springer.com/article/10.1186/1477-7525-7-43 (psychometric study)
And as you can see, not all the time both studies are made for the same group. Once the cross-cultural study is done, the scale can be validated for any reference group in the target country. The same is not true for the psychometric study: it only works for the reference sample.
In sum, publishing these different studies in different articles ables to (1) reach the proper audience (Brazilian for the cross-cultural; Brazilian and foreign for the psychometric study); (2) provide the proper methodological details for each of then; (3) present the results in details; (4) avoid multiple target language versions of the same scale/inventory/instrument (since enough information was provided in the article dedicated to it, collaborating for a “homogeneity” of the instrument, which ables to compare data collected in different times and/or different sample.
However, despite we strongly supports our point of view, we will concider your argumentation for future publications
Q2.:The arguments provided by the authors regarding the population they have used to assess the questionnaire seem correct to me, but I continue to maintain that it is necessary to include the target population to which the questionnaire will be directed. In fact, it would add more value to the item right from the start.
Answer: Dear colleague, we agree with you, and in fact, we gave this information right from the start, adding in our title that the measure was being validated “for people living with HIV / AIDS”. However, in order to be clear for our readers, we added the information at the beginning of the methods section, under the subtitle 2.1 Study design and sample size calculation (lines 196-197).
Q3.:Lastly, I thank you for taking the time to answer all the questions. I encourage you to continue working in this much-needed area and to continue your research progress in this area.
Answer: We do thank you, for this last one and for all your comments that helped us to improve our manuscript
